# Evaluation Method for Node Importance of Urban Rail Network Considering Traffic Characteristics

**Ting Chen [1], Jianxiao Ma [1,\*], Zhenjun Zhu [1] and Xiucheng Guo [2]**

1. School of Automobile and Traffic Engineering, Nanjing Forestry University, Nanjing 210037, China
2. School of Transportation, Southeast University, Nanjing 210096, China

\* Correspondence: majx@njfu.edu.cn

**Abstract:** As a sustainable means of public transport, the safety of the urban rail transit is a significant section of public safety and is highly important in urban sustainable development. Research on the importance of urban rail stations plays an important role in improving the reliability of urban rail networks. This paper proposed an improved method for evaluating the importance of urban rail stations in a topology network, which was used to identify the key stations that affect the urban rail network performance. This method was based on complex network theory, considering the traffic characteristics of the urban rail network that runs on specific lines and integrating the structural characteristics and interrelationship of the lines where the stations are located. Hereafter, this method will be abbreviated as CLI. In order to verify that the high importance stations evaluated by this method were the key stations that had a great impact on the urban rail network performance, this paper designed a comparative attack experiment of betweenness centrality and CLI. The experiment was carried out by taking the Suzhou Rail Transit (SZRT) network as an example and the largest connected subgraph as well as the network efficiency as indicators to measure the network performance. The results showed that CLI had a greater impact on network performance and could better evaluate the key stations in the urban rail network than node degree and betweenness centrality.

**Keywords:** urban rail network; complex network; topological structure; node importance; network performance

## 1. Introduction

As a component of urban infrastructure, urban rail transit is increasingly used due to its energy-saving, environmentally friendly, high-efficiency, and other characteristics. In China, more and more cities are building urban rail networks. According to the statistics of the Ministry of Transport, by 2022, a total of 281 urban rail lines and 9246 km of running mileage were opened in 51 cities in China, of which 31 cities had more than three urban rail lines. Moreover, the number of unban rail construction is still growing at a high speed. In 2021, 35 new urban rail lines and 1168 km of new running mileage were built, which was an increase of about 15% over 2020. At present, China not only has many cities with rail transit networks and a rapid pace of construction, but also an increasing number of passengers. In 2021, the annual passenger volume increased by 6.12 billion person–time, about 35%, compared with 2020, reaching 99.2% in 2019 [1]. These data demonstrate that urban rail transit is gradually becoming one of the most important public transport tools in cities, and more and more passengers rely on the travel mode by urban rail transit.

Due to the large traffic volume and high passenger density of urban rail transit, once a dangerous situation occurs, it will cause huge economic losses, pose a huge threat to the life safety of passengers, and affect the service level. Therefore, it is necessary to ensure the safe operation of urban rail transit, improve its network performance, and ensure that it has enough reliability to not collapse in the face of threats. With the expansion of the urban rail transit network, it will be more seriously affected when faced with interference,

such as terrorist attacks, extreme weather events, congestion, cybersecurity attacks, and electricity shut offs, which will not only damage the structure of the urban rail network but also reduce the service level of the urban rail transit. On 2 August 2021, two urban rail trains collided in Boston, USA, causing 25 people to be injured, and the station and section were closed for maintenance for a long time [2]. On 20 July 2021, Zhengzhou was hit by extreme torrential rain, and Line 5 of urban rail transit was seriously flooded, resulting in the death of 14 passengers and the shutdown of the whole network [3]. It can be found from these events that although the interference occurs in individual stations or sections, it will affect the operation of a line and even the entire rail network.

With the network operation of urban rail transit, the connection between stations is closer, and the interaction between lines is more significant. From the perspective of network structure, the failure of any station in the network will reduce the network connectivity and network efficiency, thereby reducing the service level of urban rail transit, making passengers crowded and stranded, and even affecting the safety of passengers. There are some stations in the network that will have a greater impact on the network structure when they are disturbed [4,5]. If we can find these key stations, we will focus on their maintenance and recovery, which will improve the anti-interference ability and stability of the structural network and reduce the impact of interference events on the network structure. This article will determine how to find these key stations. In complex network theory, the importance of nodes is described by the index of node importance [6–8]. The higher the node importance, the greater the impact of the node on the network performance. The nodes with the highest node importance ranking are the key nodes [9,10]. This paper aims to build a more accurate evaluation method of node importance and carry out simulation experiments to prove that this method is more suitable for urban rail transit networks than other methods.

The paper is organized as follows: Section 2 reviews the literature on the evaluation of the node importance of complex networks and urban rail networks. Section 3 introduces the topology analysis of the urban rail network, typical indicators of complex networks, and an index considering the characteristic of specific lines in urban rail networks. Section 4 gives the improved evaluation method for the node importance of urban rail networks and verifies it. Section 5 takes the SZRT network as a case for study. Section 6 summarizes the study conclusions and suggests future areas of study.

## 2. Literature Review

In recent years, using complex network theory to solve real network problems has been a research hotspot, and the evaluation of node importance is a very popular research topic in complex networks [11–15]. The application of complex network theory to the urban rail network has promoted our understanding and control of the urban rail network. Research on the evaluation of the node importance of urban rail networks is the basis for maintaining the safety of rail transit networks and ensuring the service level of network operation. In this section, we review relevant literature and introduce relevant techniques from the following two aspects.

### 2.1. Evaluation of Node Importance of Complex Networks

Research on the node importance of complex networks can be defined as research on the influence of nodes on the complex networks. The higher the node importance, the greater the influence of nodes on the network. The evaluation of node importance refers to the process of measuring and ranking the influence of nodes on complex networks. The evaluation methods for node importance can be divided into local methods and global methods. Node degree [11] is one of the simplest and classic indicators in the evaluation of node importance, which is simple to calculate and only considers local information. Kitsak et al. [16] proposed k-shell decomposition to find nodes at the core of the network, which is an extended algorithm based on the node degree. The granularity of indicators based on node degree is coarse, and there may be many nodes in the same importance

degree. Lü et al. [17] introduced the H-index algorithm to further divide node importance, and the H-index was initially used to evaluate academic influence. The node importance evaluation methods considering the global attributes of nodes mainly include betweenness centrality, closeness centrality, etc. The betweenness centrality refers to the proportion of the shortest path through a node to all the shortest paths in the network, which reflects the node's control over the entire network of information dissemination [18]. The closeness centrality considers that the node closest to all nodes in the network is important, that is, the shorter the distance between the nodes, the faster the propagation rate [19]. In view of the defects of a single index, some scholars improved the evaluation method for node importance via multi-index fusion. Li et al. [20] proposed an evaluation method for node importance by combining the node degree, relative entropy, and TOPSIS comprehensive evaluation method.

### 2.2. Evaluation of Node Importance of Urban Rail Networks

The famous Königsberg's Bridge Problem, solved by Euler in 1735, is considered to be a simplified and the earliest transportation optimization problem. It not only laid the foundation for graph theory, but also creatively proposes the use of network topology to solve traffic problems. With the development of graph theory, network analysis has gradually become the most effective method for traffic research [21,22]. Furthermore, the emergence of complex networks provides a systematic theoretical method for solving traffic network problems. Lin et al. [23] comprehensively introduced the application of complex network theory in the field of transportation. As a part of the transportation system, the URTN has obvious complex network characteristics such as small-world property, scale free property, etc. [24,25]. Therefore, using complex network theory to study urban rail networks has become a mainstream method and hotspot. Many studies on the vulnerability and robustness of subway systems are based on complex network theory [26]. Zhang et al. [27] studied the networked characteristics of the Shanghai urban rail network and analyzed its robustness and reliability based on complex network theory. On this basis, Zhang et al. [28] studied the topological characteristics, found many similar characteristics among urban rail networks around the world, and discussed the failures to discuss the vulnerability of the urban rail network. Therefore, it has been proved that the network performance of URTNs can be studied with complex network theory.

Research on network performance found that there are some nodes that play a vital role in the network performance [29–31]. The early methods used to screen these key nodes included node degree and the betweenness centrality. It was found that intentional attacks based on node degree and the betweenness centrality can cause network collapse faster than random attacks [32–34]. Sun et al. [32] proved that compared with being attracted randomly, urban rail networks are more vulnerable when attacked by the highest node degree and betweenness centrality stations, and the stations with the largest node degree have a greater impact on the network scale, while the stations with the highest betweenness centrality have a greater impact on the network efficiency. It is concluded that different node importance evaluation indicators have different effects on the rail network performance.

Some scholars try to apply and compare the existing complex network importance evaluation methods to the URTNs [35,36]. Lai et al. [36] established urban rail topology models of multiple cities, the toke network efficiency, and the largest connected subgraph as indicators to measure the network performance, and they found that the KSD identification method was superior to the node degree, neighbor node degree, DKS, and DKSN identification methods. Some scholars realized that there are differences in node importance in different application environments, and they tried to improve and even build new evaluation methods to more accurately obtain the key nodes that have the greatest impact on the urban rail network performance [37–39]. Liu et al. [37] defined the topology of urban rail networks using the idea of Space P and Space L and a new parameter to evaluate the node importance and found that urban rail networks are more vulnerable to intentional attacks than random network attacks based on this node importance method.

Xia et al. [39] proposed a node importance evaluation method (SIRank) and compared SIRank with traditional methods by using the experiment of attacking nodes with the highest importance. For the directed weighted network and the undirected unweighted network, the results obtained by using the same node importance evaluation method were different. Du et al. [40] offered a new method of node importance, which was the Improved Topological Potential model considering Entropy (ITPE), and found that the node importance evaluation methods applicable to the directed and weighted networks was not applicable to the unweighted and undirected topological networks.

From the perspective of the development of evaluation methods for the node importance of complex networks, it is a development trend to integrate local and global indicators and build comprehensive indicators with comprehensive attributes, which can more comprehensively evaluate the impact of nodes on network performance [41]. However, few evaluation methods for node importance applied to urban rail networks consider the global and local attributes of indicators. For different analysis needs, new evaluation methods for node importance are constantly emerging, and the application accuracy of evaluation methods for node importance in urban rail networks still has room for improvement [42,43]. The urban rail transit network is different from other networks in that it runs on specific lines [44]. Although researchers have realized that urban rail networks are different from other complex networks, they have not integrated the characteristics of urban rail transit that runs on specific lines into the evaluation of the node importance of urban rail transit networks.

In order to solve the above problems, this paper fully considers the traffic attributes of urban rail networks and will construct an evaluation method for node importance suitable for structural networks of urban rail transit. On the basis of traditional node importance indicators of complex networks, we integrate indicators with local and global attributes to make the impact of this method on network performance more comprehensive. First, we used Space L to build the structure network of urban rail transit. Then, considering the characteristics of the urban rail network running on specific lines, we used Space Syntax to obtain the operation accessibility of the nodes. Finally, we obtained the evaluation method for node importance that integrated global and local attributes and traffic attributes. The biggest difference between this evaluation method for node importance and other methods is that it considers the characteristics of the urban rail network that runs on specific lines, so it is named with the abbreviation CLI, meaning "considering line importance". Taking the SZRT network as an example, the effectiveness and accuracy of CLI was verified.

## 3. Methodology

In order to build an evaluation method for node importance with higher accuracy and that is more suitable for urban rail networks, this paper improved the typical evaluation method for the node importance of complex networks and fully considered the global and local attributes of the indicators. In addition, in order to make the evaluation method for node importance more applicable to the urban rail network, this paper considered the characteristics of the urban rail transit that runs on specific lines, and the results obtained conformed to the operation status of the urban rail networks. The methodology herein is shown in Figure 1 and consists of the following steps:

(1)   Constructing the urban rail network with the Space L model;
(2)   Calculating the node degree and betweenness centrality of the nodes in the Space L network, based on complex network theory;
(3)   Analyzing the relationship between the lines where the stations are located with Space Syntax and obtaining the operation accessibility of the nodes;
(4)   Building a CLI node importance method that considers the node degree, betweenness centrality, and operation accessibility of the nodes;
(5)   Ranking the stations according to the calculation results of the node degree, betweenness centrality, and CLI, and comparing the stations with high importance under each method;

(6)    Attacking high-betweenness-centrality and high-CLI nodes in turn, comparing the network performance degradation under different node failure scenarios and giving the best method.

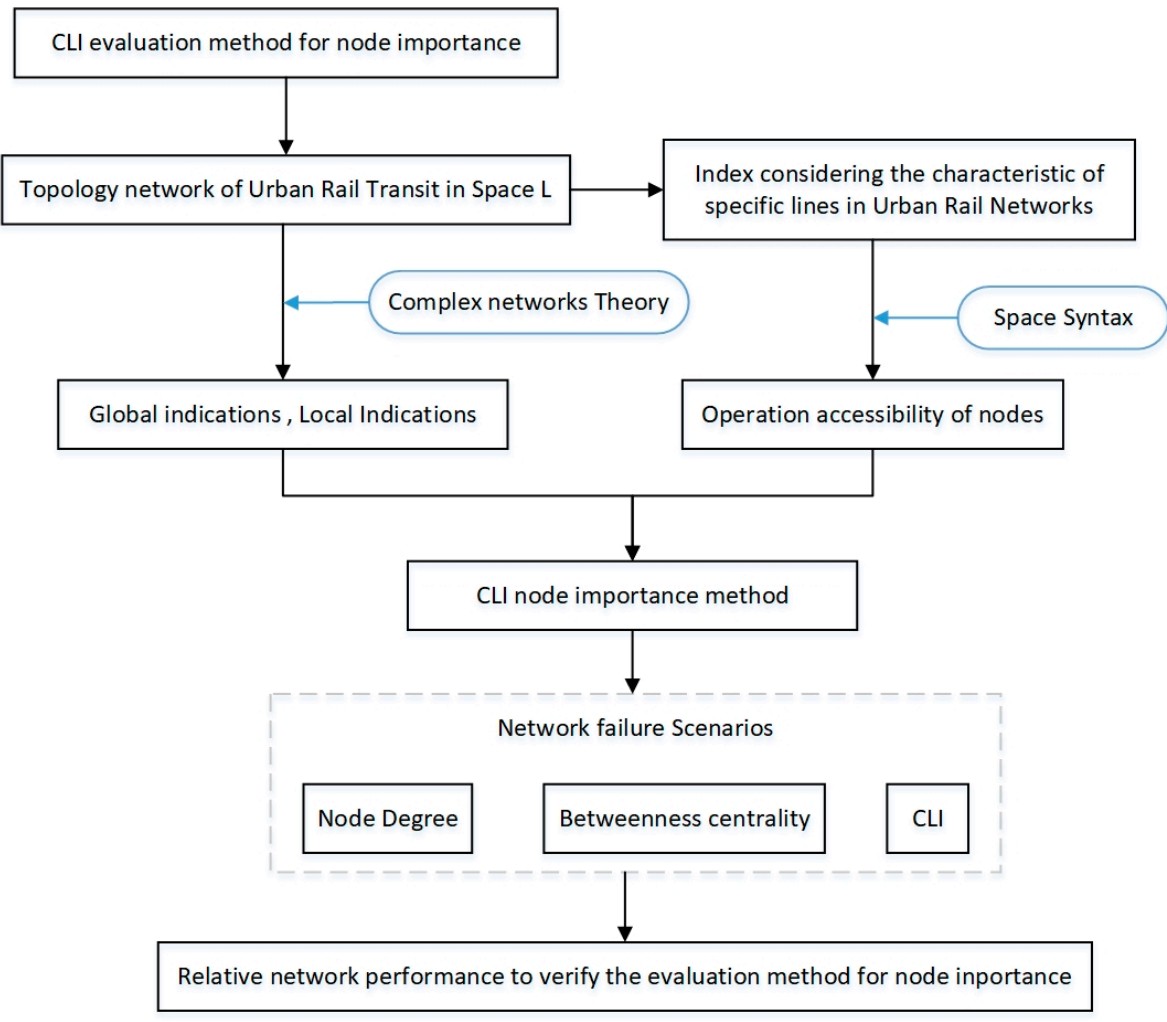

**Figure 1.** Methodology of the CLI evaluation method for node importance.

## 4. Urban Rail Complex Networks

### 4.1. Topology Analysis of Urban Rail Networks

There are four main methods to construct the topology structure of transport networks based on complex network theory, namely Space L, Space P, Space B, and Space C, which were introduced in detail in the literature [42,45]. Space L and Space P models are commonly used in urban rail networks. Figure 2 shows the results of constructs of the same simple urban rail network with these two models. In the Space L model, the stations are regarded as nodes, and the link between two nodes indicates that there is at least one route that services the two consecutive stations and no other intermediate stations between them. In the Space P model, the stations are regarded as nodes, and if any two stations are serviced by at least one common route, there is a link between the two stations. The Space L model is the most intuitive expression of the urban rail network, which is mainly used to study the topology structure characteristics and vulnerability of the rail transit network; in the Space P model, the neighbors of a node are all stations that can be reached without changing the lines, so the Space P model is usually used to study the transfer characteristics of the urban rail network.

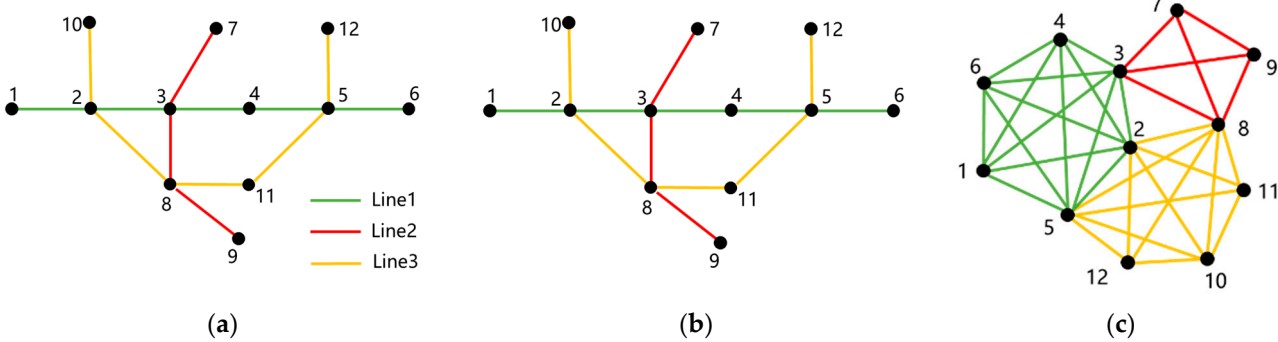

**Figure 2.** (**a**) An urban rail network. (**b**) Space L graph. (**c**) Space P graph.

It can be seen from Figure 2 that the topology network constructed by Space L was more similar to the urban rail network in reality. This paper studies the topological feature of node importance in the urban rail network. Stations correspond to nodes in the topology network. Whether the stations are directly connected indicates the nodes' connection relationship in the topology network. Therefore, Space L is more suitable for constructing a topology network of urban rail transit. According to complex network theory, the urban rail network can be represented as a matrix $A_{ij}$, which is a square matrix used to represent a finite network. A network with $n$ nodes can be represented as an adjacency matrix table with $n \times n$ elements, and element $a_{ij}$ represents the connection between node $i$ and node $j$:

$$a_{i,j} = \begin{cases} 1, & \text{if there is a link between node } i \text{ and node } j, \\ 0, & \text{otherwise.} \end{cases} \tag{1}$$

### 4.2. Typical Indicators of Complex Networks

In an urban rail topological network constructed by the Space L model, the typical indicators representing node importance are node degree (D), betweenness centrality (BC), closeness centrality (CC), and eigenvector centrality (EC). The definitions and calculation formulas for the indicators are showed in Table 1 [22].

**Table 1.** The definitions and calculations of typical node importance indicators.

| Index | Definition | Formula |
|---|---|---|
| D | D measures the number of other nodes directly connected to a node. | $D_i = \sum_{j=1}^{n} a_{ij}$ <br> $n$ is the total number of network nodes. |
| BC | BC measures shortest number of paths through a node. | $BC_i = \sum_{s,j \in V} \frac{\sigma(s,j|i)}{\sigma(s,j)}$ <br> $\sigma(s,j|i)$ represents the number of all the shortest paths through node $i$ in the shortest path from node $s$ to node $j$. |
| CC | CC measures the ability of a station to affect another node through the network. | $CC_i = \left[ \sum_{j}^{n} d_{ij} \right]^{-1}$ <br> $d_{ij}$ represents the shortest distance between node $i$ and node $j$. |
| EC | EC measures the shortest number of paths through a node. | $EC_i = \frac{1}{\lambda} \sum_{j \in \Gamma(i)}^{n} a_{ij} x_j$ <br> $\Gamma(i)$ is the set of neighbor nodes of node $i$, where $\lambda$ is a constant. |

D directly reflects the position of a node in the network or the relationship with adjacent nodes, taking into account the local attributes of the node. BC gives the number of shortest paths that pass through a node in the network, which can best measure the

connectivity potential of the station in the urban rail network. The larger the BC, the more times the station passes through the shortest path in the network. CC underlines the position of a node in the network, and CC is closer to the geometric center of the network. BC and CC reflect the global characteristics of the nodes in complex networks.

### 4.3. Index Considering the Characteristic of Specific Lines in Urban Rail Networks

When analyzing the urban rail topological network, in addition to fully considering the complex network characteristics, it is also necessary to combine the urban rail network operation characteristics. In actual operation, urban rail transit trains operate on dedicated lines. Trains on different lines generally do not participate in the operation of other lines, so the relationship between the lines should not be ignored.

As shown in Figure 2, station 1 to 3 and station 1 to 8 had the same convenience in complex networks structured by Space L. However, when actually taking urban rail transit, passengers need to get off at station 2 and transfer from Line 1 to Line 3 before arriving at station 8. Therefore, when studying the station importance in urban rail networks, on the basis of typical methods for the node importance of complex network, the relationship between the lines where the stations are located should be considered. This paper proposes the operation accessibility of stations, which is equal to the accessibility value of the line where the station is located, as shown in Formula (2).

$$SA_i = LA_p \tag{2}$$

where $SA_i$ is the operation accessibility of station $i$ and $LA_p$ is the accessibility value of line $p$.

Space Syntax [46] focuses on the organizational structure, arrangement, and connection order between spaces. We used the configuration relationship to analyze the elements in the space (field of view, point, line, convex space), and Figure 3 takes a room plan as an example:

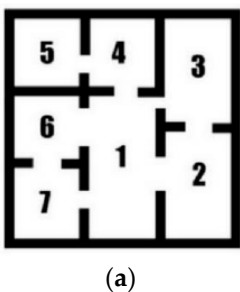 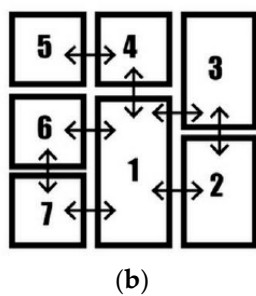 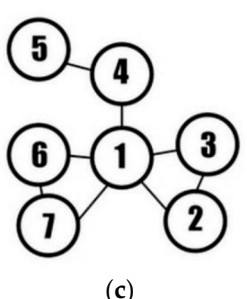 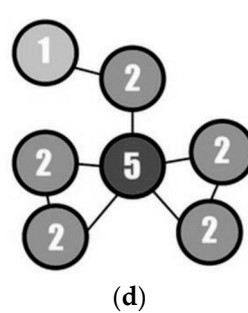

(a) (b) (c) (d)

**Figure 3.** Schematic diagram of Space Syntax. (**a**) Room plan; (**b**) divide convex space; (**c**) abstract into graph theory; (**d**) quantitative analysis connection.

We analyzed the line accessibility with Space Syntax, abstracted the urban rail line into points, and calculated the line accessibility $LA_p$ as follows:

$$LA_p = \frac{D_l}{RA_p} \tag{3}$$

$$D_l = \frac{2\left\{l\left[log_2\left(\frac{l+2}{3}-1\right)+1\right]\right\}}{(l-1)(l-2)} \tag{4}$$

$$RA_p = \frac{2(MD_p-1)}{l-2} \tag{5}$$

The links between adjacent lines were weighted. If the number of intersections of two intersecting lines is $k$, the distance weight between the intersecting lines is $1/k$. After

the traditional distance value of the mean depth value $MD_p$ is improved to the distance weight value, the $MD_p$ and the shortest distance $d_{pq}$ are calculated as follows:

$$MD_p = \frac{\sum_{q=1}^{l} d_{pq}}{l - 2} \tag{6}$$

$$d_{pq} = \sum_{1}^{g} \frac{1}{k_g} \tag{7}$$

where $p, q$ are nodes in the Space Syntax network, representing two lines in the orbital topology network; $g$ is the link segment connected between nodes $p$ and $q$, representing the transfer station in the urban rail network; $1/k_g$ is the weight of the $g$-th link between nodes $p$ and $q$; $l$ is the number of nodes in the Space Syntax network, representing the number of lines in the urban rail network; and $D_l$ is the value to eliminate the influence of the topology connection.

## 5. Node Importance of Urban Rail Networks

### 5.1. Improved Evaluation Method for Node Importance

Section 4.2 introduces the typical indicators used for the node importance analysis of complex networks and divides these indicators into local indicators and global indicators. In order to give consideration to the two attributes, the node importance evaluation method in this paper considered the use of both the node degree and betweenness centrality. On this basis, considering the traffic characteristics of the urban rail network that runs on specific lines, the node importance index CLI of the urban rail topological network is constructed as:

$$CLI_i = w_1 \widetilde{D_l} + w_2 \widetilde{BC_l} + w_3 \widetilde{SA_l} \tag{8}$$

where $\widetilde{D_l}$ is the standardized node degree; $\widetilde{BC_l}$ is the standardized betweenness centrality; $\widetilde{SA_l}$ is the standardized station accessibility; $w_1, w_2, w_3$ are weight coefficients of the node degree, betweenness centrality, and station accessibility, respectively.

The standardized formula of node degree is as follows:

$$\widetilde{D_l} = \frac{D_i - D_{min}}{D_{max} - D_{min}} \tag{9}$$

where $D_{min}$ is the minimum value of the node degree without 0–1 standardization and $D_{max}$ is the maximum value of the node degree without 0–1 standardization. The standardized formula of $BC$ and $SA$ are the same as above.

$w_1, w_2, w_3$ are calculated by the Entropy Method, and the Entropy Method will not be introduced here in detail.

### 5.2. Method Validation

The node importance evaluation method was used to find the station that had the greatest impact on the urban rail network, while the process and degree of the urban rail network affected cannot be directly judged, so this paper measured the impact of node failure under attack on the network performance. Here, for the attack strategies, we selected the typical index betweenness centrality and the CLI index proposed in this paper as the basis for comparison, and for the network performance, we selected the largest connected subgraph and network efficiency. The stations of the urban rail network were sorted from large to small according to the value of BC and CLI, and the stations under the two sorts were attacked, respectively, to obtain the decline in the largest connected subgraph and

network efficiency of the network, so as to verify the applicability of the importance index CLI proposed in this paper. The largest connected subgraph is shown in Formula (10).

$$M = \frac{N_i}{N_0} \tag{10}$$

where $N_0$ is the number of nodes in the initial network and $N_i$ is the number of nodes in the largest connected subgraph after the failure of station $i$. The largest connected subgraph reflects the transport functional integrity of the remaining network after the failure of individual nodes. The smaller the largest connected subgraph, the greater the impact of the failed node on the network performance, and the more important the node is.

The formula for calculating network efficiency is as follows:

$$E = \frac{1}{C_N^2} \sum_{i>j}^{N} \frac{1}{d_{ij}} \tag{11}$$

where $d_{ij}$ is the shortest path length between nodes $i$ and $j$. The network efficiency reflects the connectivity between nodes in the network. The smaller the network efficiency is, the greater the impact of the failed node on the network performance is, and the more important the node is.

## 6. Experimental Analysis and Discussion

### 6.1. Typical Characteristic Indicators of Suzhou Railway Network

This paper takes the Suzhou Rail Transit (SZRT) network in 2022 as an example. All the data are from Suzhou Rail Transit Group Co., Ltd. The SZRT network consists of six operation lines, with 154 stations in total. The numbers of stations are shown in Table 2, and the connection relationships between the stations are shown in Table 3. We input the information in Tables 2 and 3 in Gephi to obtain the topological structure of the network of SZRT, as shown in Figure 4.

**Table 2.** Node data.

| ID | Label | Line | ID | Label | Line | ID | Label | Line |
|----|-------|------|----|-------|------|----|-------|------|
| 1 | 140 | 1 | 59 | 340 | 3 | 120 | 764 | 4F |
| 2 | 141 | 1 | 60 | 341 | 3 | 121 | 765 | 4F |
| 3 | 142 | 1 | 61 | 342 | 3 | 122 | 766 | 4F |
| … | … | … | … | … | … | … | … | … |
| 25 | 238 | 2 | 93 | 440 | 4 | 127 | 521 | 5 |
| 26 | 239 | 2 | 94 | 441 | 4 | 128 | 522 | 5 |
| 27 | 240 | 2 | 95 | 442 | 4 | 129 | 523 | 5 |
| … | … | … | … | … | … | … | … | … |

**Table 3.** Link data.

| Source | Target | Weight | Source | Target | Weight |
|--------|--------|--------|--------|--------|--------|
| 1 | 2 | 1 | … | … | … |
| 2 | 1 | 1 | 90 | 153 | 1 |
| 2 | 3 | 1 | 90 | 154 | 1 |
| … | … | … | 154 | 90 | 1 |

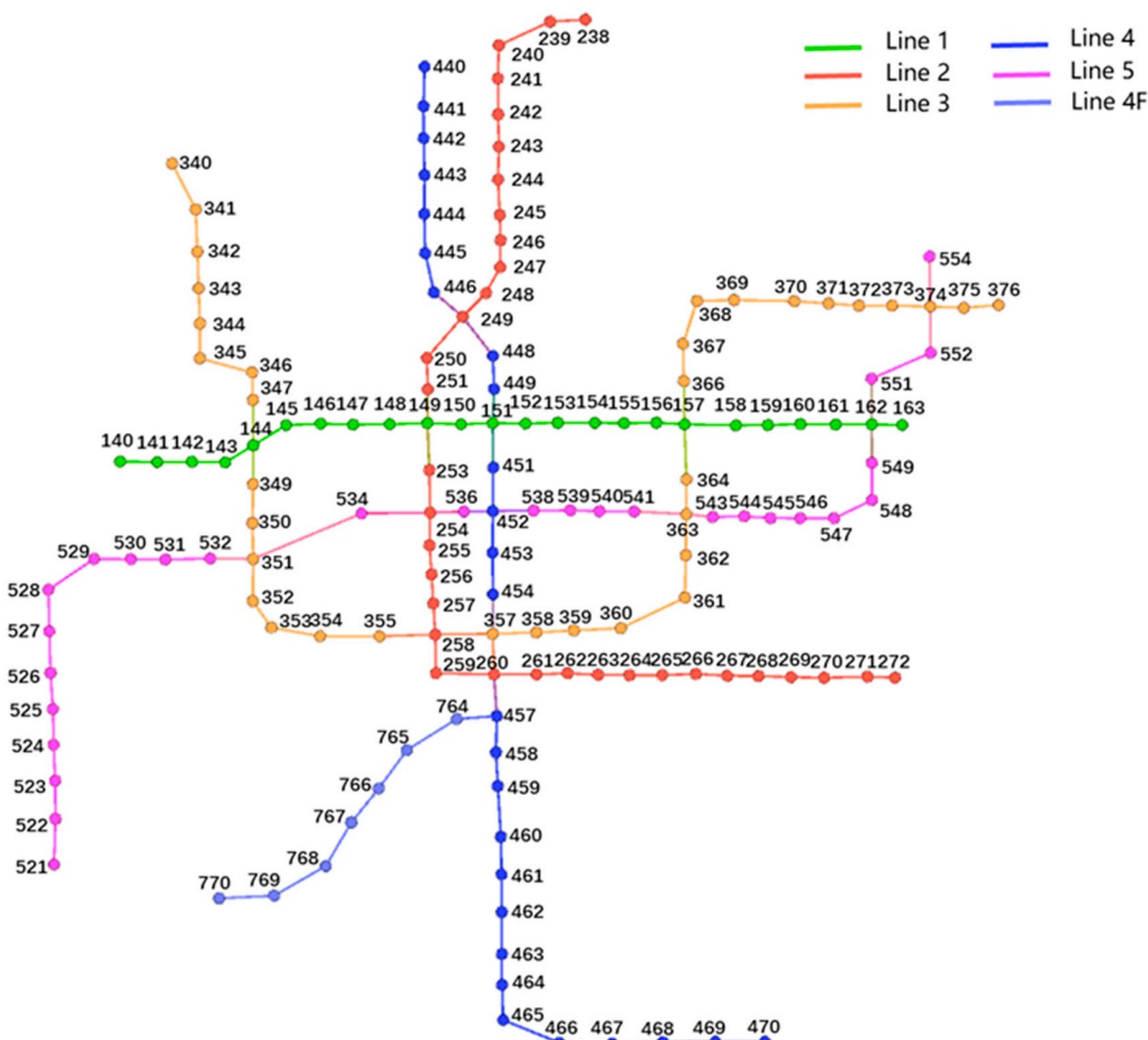

**Figure 4.** SZRT topology network.

The basic parameters of the SZRT topology network constructed by the Space L model are shown in Table 4. It can be seen that the average node degree of the SZRT network was 2.117, which is consistent with the structural characteristics of the urban rail network. In the SZRT network, except for the transfer nodes, most nodes had a degree of two, and a few edge stations had a degree of one. The network diameter represents the maximum distance of all node pairs in the network, and the maximum distance between two stations of the SZRT network is 34. The average path length represents the average distance of all the node pairs in the network. The average path length of the Suzhou Rail Transit network was 13.718. The cluster coefficient describes the possibility that individual neighbor nodes in the network are also neighbors to each other. The network cluster coefficient is the average of the cluster coefficients of all nodes in the network. The network cluster coefficient was 0, indicating that the SZRT network did not have the small-world properties.

**Table 4.** Basic parameters of topology network.

| Line Number | Node Number | Link Number | Average Node Degree | Network Diameter | Average Path Length | Network Cluster Coefficient |
|---|---|---|---|---|---|---|
| 6 | 154 | 163 | 2.117 | 34 | 13.718 | 0 |

We calculated the stations' node degrees and betweenness centrality of the SZRT network. The top 10 stations are shown in Table 5.

**Table 5.** SZRT network sorted by node degree and betweenness centrality of the stations.

| Order | Highest D Stations | D | Highest BC Stations | BC |
|---|---|---|---|---|
| 1 | Shihudong RD. | 4 | Shihudong RD. | 0.183 |
| 2 | Baodai RD. | 4 | Baodai RD. | 0.169 |
| 3 | Nanmen | 4 | Nanmen | 0.134 |
| 4 | Suoshanqiaoxi | 4 | Suoshanqiaoxi | 0.130 |
| 5 | Leqiao | 4 | Leqiao | 0.128 |
| 6 | Dongfangzhimen | 4 | Dongfangzhimen | 0.120 |
| 7 | Jinsheqiao | 4 | Hongzhuang | 0.118 |
| 8 | Suzhou railway station | 4 | Jinsheqiao | 0.115 |
| 9 | Laodong RD. | 4 | Suzhou railway station | 0.110 |
| 10 | Guangjinan RD. | 4 | Laodong RD. | 0.094 |

It can be seen from Table 5 that the ranking of the node degree and betweenness centrality was basically the same, and there was a big difference in Hongzhuang Station, mainly because the two indicators focused on different aspects. The node degree represents the number of links connected to the node, which is a local indicator. The betweenness centrality represents the intermediary role of the node in the network, which is a global indicator. Both of them can partly reflect the importance of the stations in the network. Like Hongzhuang Station, although there are only three adjacent nodes, it is very important for Line 4F, because other lines must pass through Hongzhuang Station to reach other stations on Line 4F. This further validates the CLI method, which comprehensively considers the rationality of node degree and betweenness centrality.

*6.2. CLI Node Importance Calculation*

The Space Syntax is used to construct the line relationships of the SZRT network, as shown in Figure 5.

According to the number of transfer stations between lines, the weight of the intersection lines is improved. The improved weight of intersecting lines is shown in Figure 5. According to the given weight value, we calculated the shortest distance $d_{pq}$ between line $p$ and $q$, and then we obtained the accessibility of all lines. The results are shown in Table 6.

**Table 6.** Results of line accessibility.

| Line | 1 | 2 | 3 | 4 | 4F | 5 |
|---|---|---|---|---|---|---|
| accessibility | 5.558 | 8.337 | 10.005 | 16.675 | 1.853 | 6.253 |

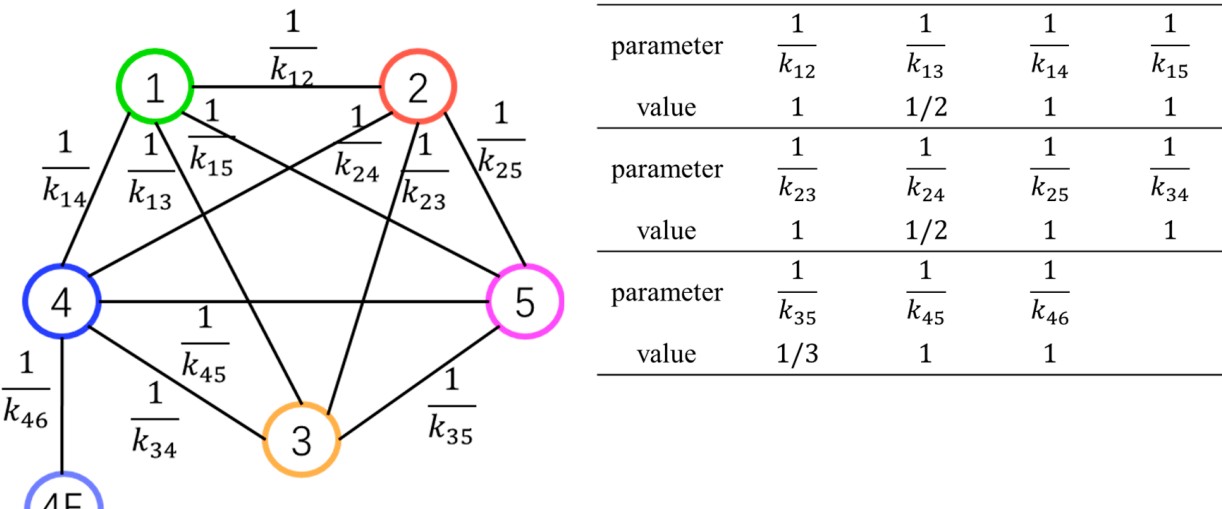

**Figure 5.** Space Syntax diagram of SZRT.

It can be seen that Line 4 of the SZRT network had the highest accessibility while Line 4F had the lowest accessibility value. This was because Line 4F only intersected Line 4, while Line 4 intersected all the other lines. Line 3 had more transfer stations with other lines, so the line accessibility value was second only to Line 4. The line accessibility calculated according to the improved Space Syntax conformed to the actual operation of the SZRT.

According to the line accessibility, the operation accessibility $SA_i$ of station $i$ was assigned. For the transfer stations, the operation accessibility of the station was the larger accessibility of the intersecting lines. Table 7 shows the operational accessibility of part of the stations.

**Table 7.** Operational accessibility of part of the stations.

| Station | Line | Operational Accessibility |
|---|---|---|
| Shihudong RD. | 2/4 | 16.675 |
| Baodai RD. | 3/4 | 16.675 |
| Nanmen | 4/5 | 16.675 |
| Suoshanqiaoxi | 3/5 | 10.005 |
| Leqiao | 1/4 | 16.675 |
| Hongzhuang | 4/4F | 16.675 |
| Dongfangzhimen | 1/3 | 10.005 |
| Guangjinan RD. | 1/2 | 10.005 |
| Xingtang ST. | 1/5 | 10.005 |
| Panli RD. | 2/3 | 10.005 |
| Xiangmen | 1 | 5.558 |
| Likou | 2 | 10.005 |
| Yinchun RD. | 3 | 10.005 |
| Huolidao | 4 | 16.675 |
| Xietang | 5 | 10.005 |
| Huagang | 4F | 1.853 |

The node degrees, betweenness centrality, and operational accessibility of the stations were obtained. The node importance CLI of the SZRT network was calculated according to Formula (8), and the CLI sorting of the stations is shown in Table 8.

**Table 8.** CLI sorting of the stations.

| Order | Highest CLI Stations | CLI |
|:-----:|:--------------------:|:---:|
| 1 | Shihudong RD. | 1.000 |
| 2 | Baodai RD. | 0.964 |
| 3 | Nanmen | 0.871 |
| 4 | Leqiao | 0.857 |
| 5 | Suzhou railway station | 0.808 |
| 6 | Suoshanqiaoxi | 0.747 |
| 7 | Hongzhuang | 0.742 |
| 8 | Dongfangzhimen | 0.721 |
| 9 | jinsheqiao | 0.708 |
| 10 | Laodong RD. | 0.623 |

The difference between the results of the CLI and node degree and betweenness centrality ranking mainly occurred in the Line 4 stations. When the characteristics of the specific line's operation are not considered, the convenience of going from a station to any station connected with the transfer station is the same. However, in the actual operation process, it is more convenient to go from a station to the station on the local line because it is not necessary to get off at the transfer station and transfer to other lines. Therefore, it is very important to consider the connection relationship between the lines to evaluate the importance of stations.

*6.3. Comparative Analysis*

In order to compare different node importance evaluation methods more objectively, attacks on the stations were arranged in order. Since the highest node degree of the stations in the SZRT network did not differ, a comparison test of the stations with a high BC and high CLI was used here. The nodes' failure processes under different cases is shown in Figure 6.

By comparison, in the attack scenarios sorted by different methods for node importance, there were a few differences between the failed nodes of the top 20 BC ranking and CLI ranking. Most of the top 20 failed nodes were transfer stations in the network. When the ranking exceeds 20, the gap between the failed nodes widens. The failed nodes of the top 20–40 BC rankings were mainly distributed in Line 1, 3, and 5, while the failed nodes of the top 20–40 CLI rankings were distributed in Line 3, 4, and 4F. The failed nodes of the top 40–60 BC rankings were still distributed in Line 1, 3, and 5, while the failed nodes of the top 40–60 CLI rankings were distributed in Line 2, 3, and 5. It can be considered that the CLI node importance method considering the characteristics of specific lines in urban rail networks could increase the importance of Line 4 and the lines with greater interactions with Line 4, and the importance of stations on the relevant lines will also increase.

In order to further explore whether the CLI node importance method considering traffic characteristics is more accurate than other methods for urban rail networks, we compared the network performance degradation under the CLI failure scenario and BC failure scenario and used the two indicators of the largest connected subgraph and network efficiency to characterize the network performance. The change in the largest connected subgraph of the network is shown in Figure 7.

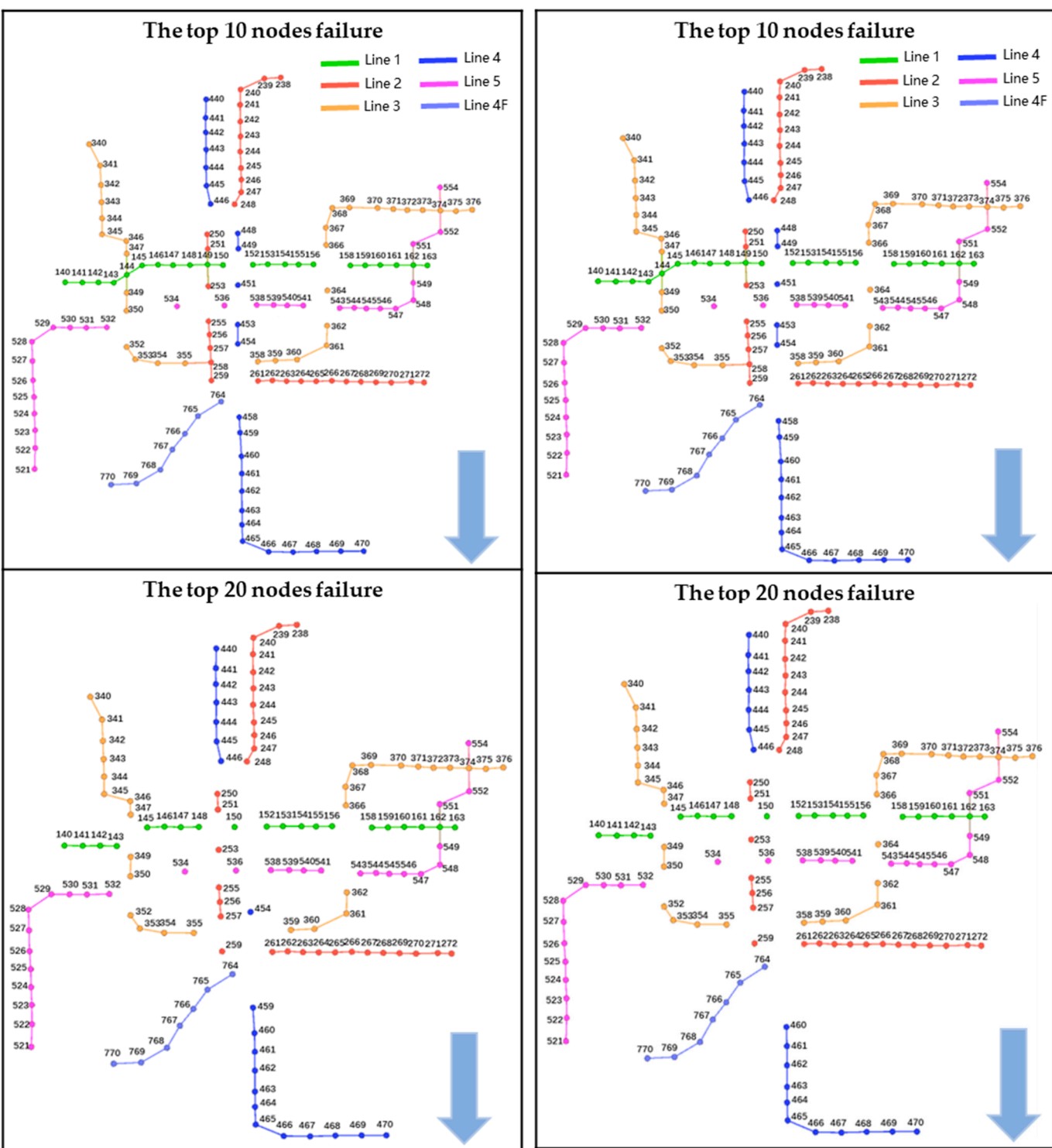

**Figure 6.** *Cont.*

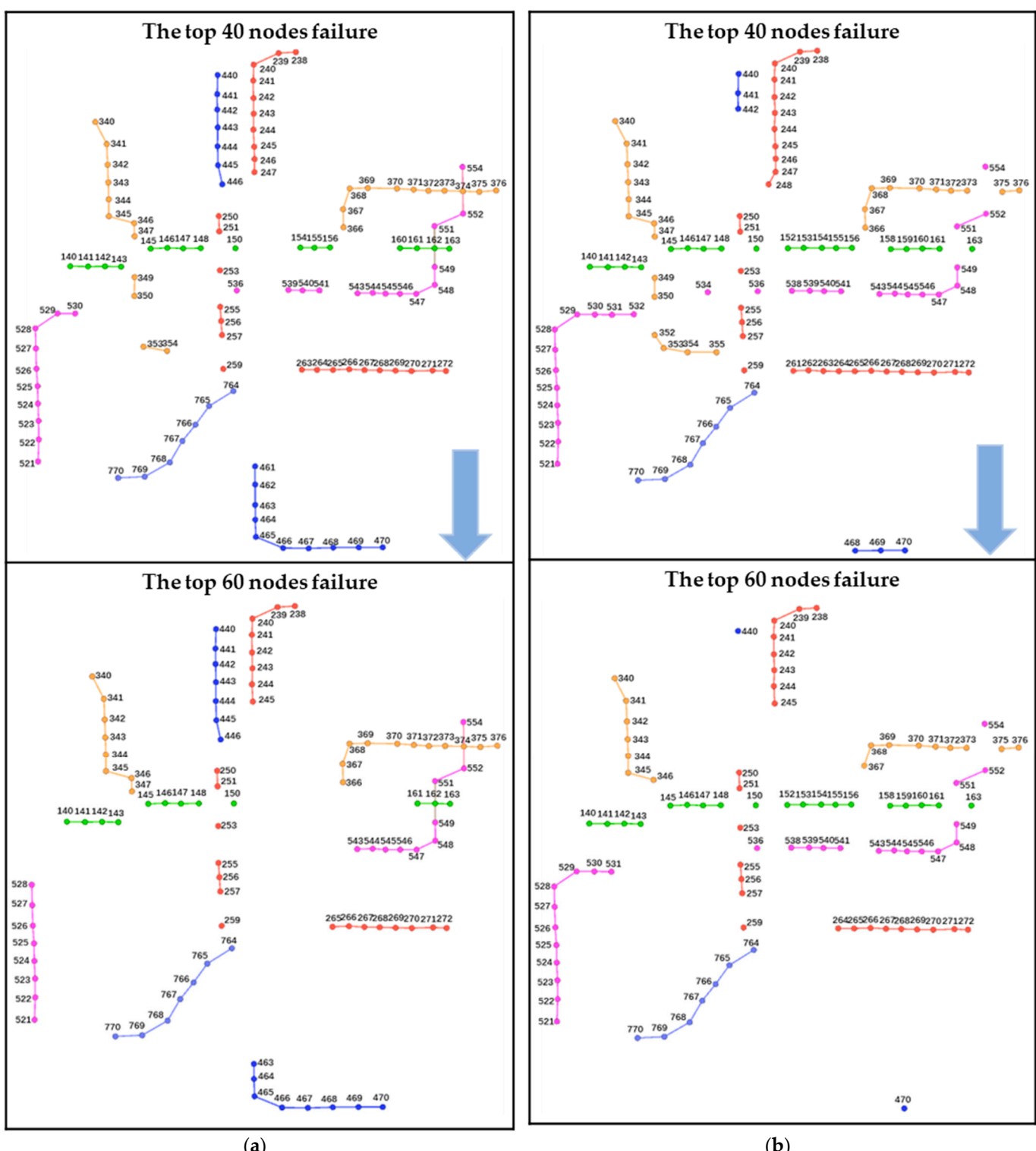

**Figure 6.** Nodes failure processes under different cases. (**a**) Case sorted by BC. (**b**) Case sort by CLI.

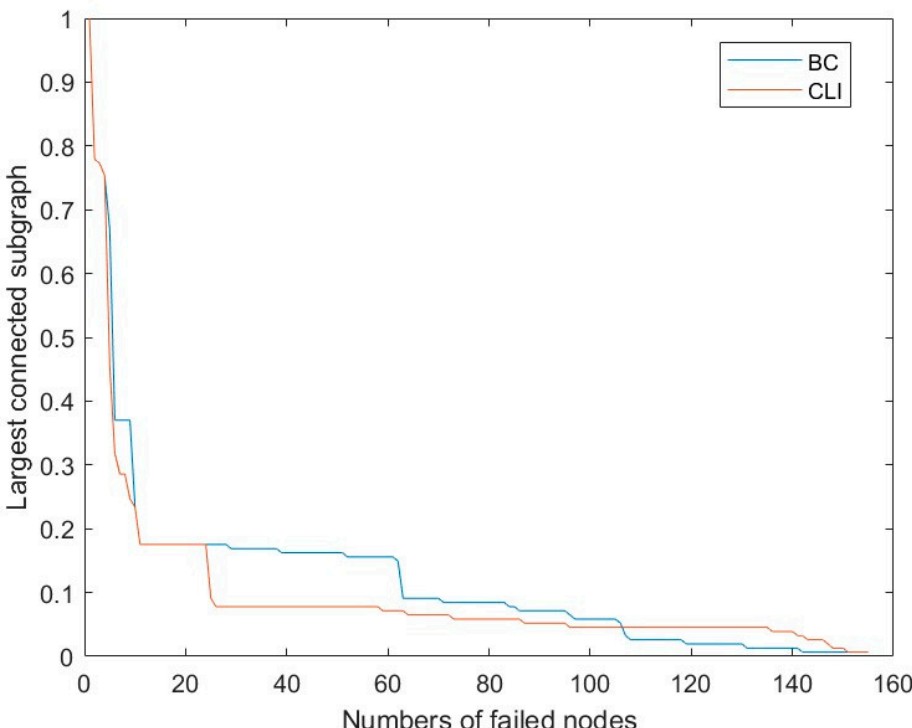

**Figure 7.** Comparison of the largest connected subgraph under different node failure cases.

It can be seen from Figure 7 that when the top five BC ranking nodes and CLI ranking nodes were attacked in turn, the largest connected subgraph of the network decreased at the same speed due to the two node failure cases. When attacking nodes in 5–10 order, the largest connected subgraph under the CLI failure case decreased significantly faster than the largest connected subgraph under the BC failure case. When attacking nodes in 10–20 order, the largest connected subgraph under the two node failure cases decreased at the same speed. When attacking nodes with a ranking greater than 20, the largest connected subgraph under the CLI failure case always dropped faster than the largest connected subgraph under the BC failure case. According to the process of node failure in Figure 6, it was found that the reasons for the gap between the two attack scenarios were mainly concentrated on the nodes of Line 4. The typical node importance method ignored the importance of the traffic characteristics of Line 4, making the node importance of Line 4 low, while the CLI method considered the traffic characteristics of each line, which improved the node importance of Line 4. The results showed that after the ranking of the node importance of the stations on Line 4 improved, the largest connected subgraph decreased faster. Therefore, it can be considered that the node failure case sorted according to the CLI made the largest connected subgraph drop faster in general, that is, the key nodes obtained by the CLI method had a greater impact on the integrity of the network than the key nodes obtained by the BC method.

It can be seen from Figure 8 that when attacking the top 20 nodes, the network efficiency under the CLI failure case was similar to that under the BC failure case. When the attack sequence was greater than 20 nodes, the network efficiency under the CLI failure case decreased faster than that under the BC failure case. In general, the node failure case sorted according to the CLI made the network efficiency drop faster, which indicated that the key nodes obtained by the CLI method had a greater impact on the transport function of the urban rail network.

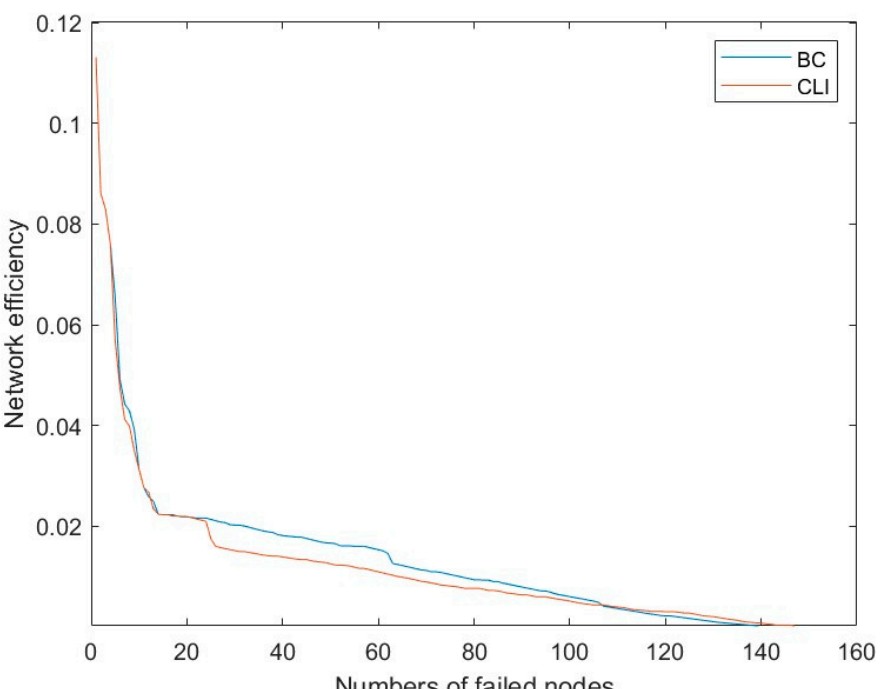

**Figure 8.** Comparison of the network efficiency under different node failure cases.

To sum up, the CLI method proposed in this paper is more accurate in evaluating the node importance of the network. Because this method considers the traffic characteristics of the line where the node is located, it is particularly suitable for urban rail transit networks. It can provide a reference for improving the service quality of urban rail transit and ensuring the safety of urban rail networks.

## 7. Conclusions

In order to more accurately evaluate the key stations of urban rail networks, using complex network theory, a station importance evaluation method considering the characteristics of operation on specific lines is proposed based on the typical node importance evaluation methods and combined with the operation characteristics of urban rail networks. The CLI evaluation method combined the local and global indicators of complex networks and the operation accessibility of nodes considering the specific lines. Take the SZRT network as an example to conduct simulation experiments, and take the node degree, betweenness centrality, and CLI as attack strategies to simulate network failure scenarios. The largest connected subgraph and network efficiency were used to measure the network performance under the network failure scenarios. The following conclusions were drawn from the simulation results:

(1) Attacking the network according to the node sequence obtained by the CLI method makes the largest connected subgraph of the urban rail network decrease faster.
(2) Attacking the network according to the node sequence obtained by the CLI method makes the network efficiency of the urban rail network decrease faster.
(3) The node importance is related to the line accessibility, and the node importance method considering traffic characteristics can make the network performance decline faster.

To sum up the above three conclusions, it is indicated that the traffic characteristics should be considered in the node importance evaluation of urban rail networks; it is more accurate to use the CLI method to evaluate the station importance of urban rail networks than other methods such as the betweenness centrality.

The research of this paper can help the operation department find the key stations that have a great impact on the urban rail network performance and ensure the safe operation of the urban rail transit. Additionally, it can provide reference for the maintenance and

recovery of the network and help the urban rail transit operation department improve the operation management level and service level.

The urban rail network not only has a static topology network layer but also a service network layer for passenger flow. This paper still has room for further improvement. In the future, we can combine the static network layer and the dynamic network layer to study the node importance of the integrated layer network that can adapt to different operational scenarios. Different urban rail networks may have different characteristics. If the CLI node importance method is applied to different urban rail networks, the results may make us find more implications. Therefore, we will consider exploring the different characteristics of urban rail networks in different cities in future research, so as to make the research on the node importance method of urban rail networks more systematic.

**Author Contributions:** Conceptualization, T.C., J.M. and X.G.; methodology, T.C.; software, T.C.; validation, T.C.; resources, X.G.; data curation, Z.Z.; writing—original draft preparation, T.C.; writing—review and editing, J.M. and Z.Z.; funding acquisition, J.M. All authors have read and agreed to the published version of the manuscript.

**Funding:** This work was funded by the Scientific Research Foundation for Advanced Talents of Nanjing Forestry University, grant number: No. 163106041; the General Program of the Natural Science Foundation of the Jiangsu Higher Education Institutions of China, grant number: 20KJB580013; and the General Project of Philosophy and Social Science Foundation of the Jiangsu Higher Education Institutions of China, funding number: 2020SJA0125.

**Institutional Review Board Statement:** Not applicable.

**Informed Consent Statement:** Not applicable.

**Data Availability Statement:** Not applicable.

**Acknowledgments:** Authors would like to acknowledge the anonymous reviewers for their constructive comments.

**Conflicts of Interest:** The authors declare no conflict of interest.

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
