# Peer review of "Evaluation Method for Node Importance of Urban Rail Network Considering Traffic Characteristics"

_sustainability, doi:10.3390/su15043582_

Round 1

Reviewer 1 Report

Peer Review Comments

1.     The authors should try to correct the grammatical mistakes in the abstract, and some sentences are not connecting.

2.     Can the authors explore more reasons why they are using the Suzhou rail transit as a case study instead of other rail transit in China?

3.     Why are they abbreviating the method they used as CLI? This should be defined.

4.     There are so many standalone sentences in the introduction section without references.

5.     The authors should state clearly in the introduction section the research aims/objectives/motivations of their research.

6.     Figure 4 should be further enlarged for more visibility of the SZRT Topology network.

7.     The authors should provide a research methodology flowchart for this research.

8.     Table 2: how are the basic parameters of the topology network determined or evaluated?

9.     All the figures inside the table on pages 12 and 13 need to be enlarged.

10.  What are the limitations of this study?

11.  The novelty needs to be clearly stated, and what are the contributions of this research to the field of rail transportation or transportation in general.

Reviewer 2 Report

I have reviewed the paper titled “Evaluation Method for Node Importance of Urban Rail Network Considering Traffic Characteristics”, which proposes an improved method to evaluate the importance of urban rail stations in topology networks. The paper is interesting and serves to solve an important topic. However, in my opinion, the paper could benefit from restructuring and adding more information for clarity in a major revision.

1-      The methodology is mixed with the literature review, and it is not very clear. Therefore, I suggest adding a separate section for the methodology and explaining it in detail.

2-      The structure of sections 3, 4, and 5 are well suited to a case study paper but not for a journal article. I suggest either resubmitting the manuscript as a case study or adding another section after section 5 to discuss the results and compare it to existing literature with high quality and recent citations.

3-      The graphics of the graphs must be improved as they are not clear with faded colours.

Reviewer 3 Report

This paper is entitled “Evaluation Method for Node Importance of Urban Rail Network Considering Traffic Characteristics”. This paper proposes an improved method to evaluate the importance of urban rail stations in topology network, which is used to identify the key stations that affect the urban rail network performance. In this case, the idea and results of the paper are interesting but the following comments can be utilized to improve this paper in future.

Abstract: Authors must provide brief information about data collection process or data utilized for this research.

Line38-40: Authors must provide citation for all information in “Introduction” and “Literatre Review”. Therefore, this paragraph needs reference(s) from recent research.

Line60-61: “There are some stations in the network that will have a greater impact on the network structure when they are disturbed.”: if this sentence is related to the current research, it must be rewrite again. Otherwise, it needs at least a reference.

Line66-67: “We consider the nodes with high importance are the key nodes that have a greater impact on the network.”: How the authors understand it has “high importance”? It must be describe in brief here.

Line 79-81: “In recent years, using complex network theory to solve real network problems has 79 been a research hotspot, and evaluation of node importance is a very popular research in 80 complex network.” Why?

Line 116: “Ref. [12] comprehensively”: It is better that authors write the name of the authos(s) for citation [12].

Line 118: “properties [13], scale-free [14],” It is better that authors write the name of the authos(s) for citation [13] and [14].

General: Authors must provide detail information related to the data collection, type of data, and so on. There is a gap in this manuscript.

Final decision: The idea and objective of this paper are interesting. The structure of the paper is suitable. This paper is appropriate for publication after minor review. 

Round 2

Reviewer 1 Report

Dear Authors,

I am satisfied with the response to reviewer's comments

Reviewer 2 Report

Thanks to the authors to address the comments.

I think the paper in good shape for publication. 

Good luck.